# Rhodium-catalyzed enantioselective and diastereodivergent access to diaxially chiral heterocycles

Yishou Wang[1], Xiaohan Zhu[2], Deng Pan[3], Jierui Jing[2], Fen Wang[2] ✉, Ruijie Mi[1], Genping Huang [3] ✉ & Xingwei Li[1,2] ✉

N-N axially chiral biaryls represent a rarely explored class of atropisomers. Reported herein is construction of diverse classes of diaxially chiral biaryls containing N-N and C-N/C-C diaxes in distal positions in excellent enantioselectivity and diastereoselectivity. The N-N chiral axis in the products provides a handle toward solvent-driven diastereodivergence, as has been realized in the coupling of a large scope of benzamides and sterically hindered alkynes, affording diaxes in complementary diastereoselectivity. The diastereodivergence has been elucidated by computational studies which revealed that the hexafluoroisopropanol (HFIP) solvent molecule participated in an unusual manner as a solvent as well as a ligand and switched the sequence of two competing elementary steps, resulting in switch of the stereoselectivity of the alkyne insertion and inversion of the configuration of the C-C axis. Further cleavage of the N-directing group in the diaxial chiral products transforms the diastereodivergence to enantiodivergence.

Axially chiral scaffolds have gained increasing interest in the past decades owing to their profound applications as chiral ligands and oganocatalysts[1–13]. In the family of axial chirality, C−C[14–18] and C−N[19–26] axially chiral biaryls have been extensively or increasingly investigated. In contrast, N−N axially chiral biaryls remain rarely studied and their synthesis lags far behind, with the first example being disclosed by the Lu group in 2021[27]. Since then, this chemistry has attracted upsurging interest. Arguably, the rarity of N−N axially chiral biaryls is ascribed to their synthetic challenges because the relatively short N−N bond distance poses steric hindrance during the formation of new chemical bonds around this axis. Among the handful examples that have been disclosed, two synthetic strategies are followed (Scheme 1a). In most cases, an N−NH group typically undergoes organocatalyzed allylation or acylation of the NH site, resulting in size increase around the N−N axis[27–30]. Similarly, an N−NH$_2$ moiety may also undergo annulative difunctionalization to create a five-membered azacycle, as have been recently reported by the groups of Shi[31], Zhao[32], and Liu[33]. Beside such

size-increasing effect, Liu and coworkers also succeeded in desymmetrization of arenes bearing existing N−N axis[34,35].

Several underlying challenges exist behind the rarity of N−N axially chiral biaryl systems. The existing systems are restricted to those bearing a single N−N chiral axis, and no multichiral products have been explored. In line with this challenge, multi-chiral systems that integrate multi-axial[36–39] or axial and central[40–45] chirality remained rare. For asymmetric catalytic systems that generate more than one chiral element, the issue of diastereoselectivity arises in addition to enantioselectivity. Consequently, the most formidable challenge is probably stereodivergence[46] during construction of such multichiral products. In this context, diastereodivergence has been previously realized using two mutually compatible or two similar chiral catalysts in asymmetric catalysis (Fig. 1b)[46–55]. It would be ideal and highly desirable to selectively deliver different diastereomers[56–58] or even enantiomers[59–63] by employment of a sole catalyst (Fig. 1b). Nevertheless, related examples are rare and typically resulted from

[1]Institute of Molecular Science and Engineering, Institute of Frontier and Interdisciplinary Sciences, Shandong University, 266237 Qingdao, China. [2]School of Chemistry and Chemical Engineering, Shaanxi Normal University, 710062 Xi'an, China. [3]Department of Chemistry, School of Science and Tianjin Key Laboratory of Molecular Optoelectronic Sciences, Tianjin University, 300072 Tianjin, China. ✉e-mail: fenwang@snnu.edu.cn; gphuang@tju.edu.cn; lixw@snnu.edu.cn

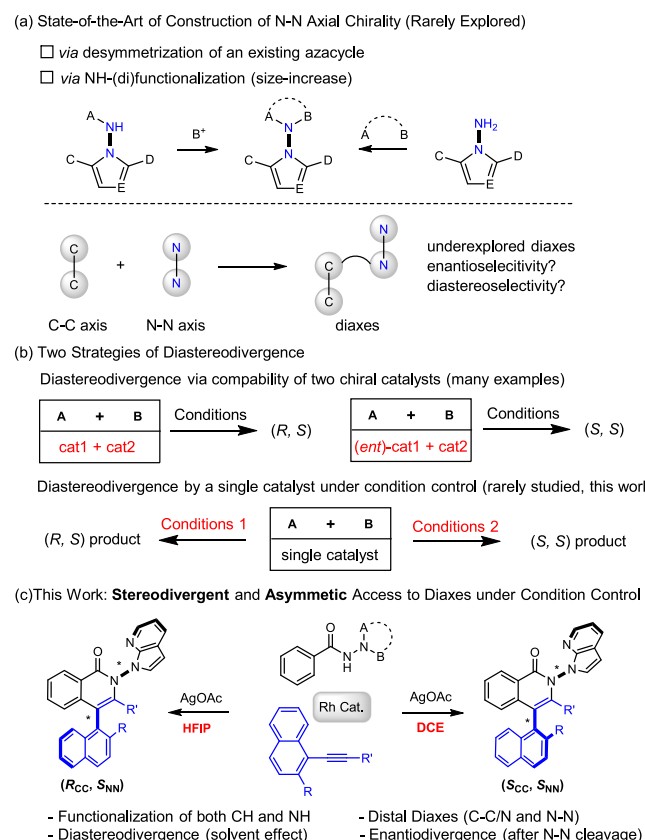

(a) State-of-the-Art of Construction of N-N Axial Chirality (Rarely Explored)
- □ *via* desymmetrization of an existing azacycle
- □ *via* NH-(di)functionalization (size-increase)

underexplored diaxes
enantioselectivity?
diastereoselectivity?

C–C axis    N–N axis    diaxes

(b) Two Strategies of Diastereodivergence

Diastereodivergence via compability of two chiral catalysts (many examples)

Diastereodivergence by a single catalyst under condition control (rarely studied, this work)

(c)This Work: **Stereodivergent** and **Asymmetric** Access to Diaxes under Condition Control

($R_{CC}$, $S_{NN}$)    ($S_{CC}$, $S_{NN}$)

- Functionalization of both CH and NH
- Diastereodivergence (solvent effect)

- Distal Diaxes (C–C/N and N–N)
- Enantiodivergence (after N–N cleavage)

**Fig. 1 | Asymmetric catalytic approaches to N–N axial chirality and diastereodivergence. a** State-of-the-art of construction of N–N axial chirality (rarely explored). **b** Two strategies of diastereodivergence. **c** Stereodivergent and asymmetric access to diaxes under condition control (this work).

accidental discovery. Besides, the N–N axial chirality mostly evolved with organocatalysis.

The rarity of N–N axial chirality calls for development of novel synthetic methods by integration with important chemistry such as C–H activation. While C–H bond activation has been applied in asymmetric construction of C–C or C–N axially chiral biaryls[64-69], including those with vicinal diaxes as reported by the groups of Zhou[70,71], Shi[72], and Niu[73], this strategy has not been applied to construction of N–N chiral axis, and stereodivergence has been even rarely involved. In this study, the N–N axial chirality and diastereodivergence are integrated by employing the N–N axis as a handle to render diastereodivergence through C–H bond activation. Our system development boils down to design of a bulky N–N containing directing group that both facilitates C–H activation and offers steric hindrance. Given abundant reports on C–H activation of benzamides[72-81], benzamide bearing a bulky N–N-directing group was designed as an arene substrate (Fig. 1c), which undergoes dynamic kinetic transformation (DKT) in the coupling with alkynes[82-85]. The oxidative coupling-annulation with a sterically hindered alkynes[82-85] is expected to create diaxes, which may offer important chemical space in addressing diastereodivergence. Despite the design, challenges in activity, enantio- and diastereoselectivity, and the long-sought stereodivergence need to be fully addressed. In this work, we report facile access to three classes of diaxially chiral biaryls containing both N–N and C–N/C axes in distal setting in excellent enantioselectivity and diastereoselectivity (Fig. 1c). Most strikingly, rarely explored diastereodivergence has been realized in the asymmetric coupling of a large scope of benzamides and 1-alkynylnaphtnalenes under condition control, and computational studies offered insightful mechanistic elucidation of the origin of diastereodivergence.

## Results

### Optimization studies

We initiated our exploration with optimization studies of the [4 + 2] annulation using a sterically hindered alkyne. To ensure catalytic activity, N-(7-azaindol-1-yl)benzamide (1) bearing a pyridine group was employed as the arene, where the pyridine ring may also facilitate the enantioselective control via chelation. A 2-substituted 1-alkynylindole (2), which often exhibited high reactivity[84-86], was selected as the coupling partner (Table 1). The coupling was performed in the presence of a chiral rhodium catalyst and Ag(I) oxidant, and the target product 3 was obtained in 70% ee and low diastereoselectivity when catalyzed by the Cramer's 2nd generation rhodium(III) catalyst[87-90] (*R*)-Rh1 with AgOAc as an oxidant at 40 °C (entry 1). Screening of catalysts returned the Rh1 as the best one in terms of activity and stereoselectivity (entries 1–4). Survey of the solvent indicated the superiority of halogenated ones (entries 5–9), and haloarene solvents were further screened. It was discovered that heavily halogenated haloarenes tended to give improved enantio- and diastereoelectivity (entries 10–13). Further variation of the additive returned 4-trifluoromethylbenzoic acid as the best additive (entries 14–16), affording 3 in excellent enantio- and diastereoselectivity even at 60 °C (entry 17).

### The reaction scope

By following the optimal conditions (Conditions A), we next examined the scope of this C–N and N–N diaxial chirality system (Fig. 2). These conditions were broadly applicable to the coupling of diverse substrates. Thus, benzamides bearing a large array of alkyl, methoxy, trifluoromethyl, trifluoromethoxy, ester, and halogen substituents at the para position all reacted smoothly with alkyne 2 (4–12, 88–99% ee), with no significant electronic effect being observed on the enantioselectivity. A similar trend was also observed when several meta-substituents were introduced (13–15). Variation of the substituent in the indole ring or in the alkyne also afforded satisfactory outcomes (16–20, 88–93% ee). The 2-sulfonyl group in the indole ring was extendable to a phosphoryl group, affording product 21 in excellent enantioselectivity (82% yield, 95% ee) using methanol as a solvent. Of note, only a single diastereomeric product (>20:1 dr) was generated in all these C–N and N–N diaxially chiral products. The absolute configuration of product 8 was determined to be (*S*, *S*) by X-ray crystallographic analysis. In addition, control experiments using an N-(1-indolyl)benzamide only gave poor reaction.

To further define the scope of arenes, a more challenging benzamide 22 bearing an open-chain N-amino group was interrogated (Fig. 2, bottom). Previously, terminal NH functionalization with no annulation has been the dominant strategy in the construction of N–N axial chirality[26-29]. Initially, its coupling with the N-alkynylindole essentially afforded no target product. To our delight, switching to a 2-substituted 1-alkynylnaphthalene[82,83] (23) as the alkyne afforded the corresponding annulation product under operationally simple conditions using AgOAc as the oxidant in MeOH (Conditions B, 24, 97% ee). In line with the above arene scope, diverse alkyl, methoxy, halogen, trifluoromethyl, and cyano groups at the para or meta positions of the arene ring were fully compatible (25–32, 92–98% ee). The reaction also tolerated different ortho substituents (33–35) albeit with lower activity for the ortho-Me group. The alkyne scope was next examined, and the presence of electron-withdrawing and -donating groups at different positions of the alkyne terminus or in the naphthalene ring only had marginal influence on the reactivity or enantioselectivity (36–47). The N-protecting group was also successfully extended to N-Cbz, and excellent enantioselectivity was realized under modified conditions (48, 96% ee). The diastereoselectivity of this reaction was generally high, and in most cases >12:1 d.r. was obtained. The absolute configuration of 47 was also determined to be (*S*, *S*) by X-ray crystallography. The most important spectroscopic feature of these products is the

**Table 1 | Optimization of reaction conditions of formation of a C–N and N–N diaxially chiral product[a]**

(R)-Rh catalyst (3 mol%)
AgSbF$_6$ (12 mol%)
AgOAc, Additive
Solvent, 40 °C, 12–72 h

(R)-Rh-1 (R = OMe, X = Cl)
(R)-Rh-2 (R = OiPr, X = Cl)
(R)-Rh-3 (R = OTIPS, X = I)
(R)-Rh-4 (R = Ph, X = Cl)

Sol-1, Sol-2, Sol-3, Sol-4

| Entry | Cat. | Additive | Solvent | Yield (%) | ee (%) | d.r. |
|---|---|---|---|---|---|---|
| 1[b] | Rh1 | AcOH | DCE | 79 | 70 | 2:1 |
| 2[b] | Rh2 | AcOH | DCE | 50 | 60 | 1.5:1 |
| 3[b] | Rh3 | AcOH | DCE | 32 | 44 | 2.5:1 |
| 4[b] | Rh4 | AcOH | DCE | 66 | 67 | 1.2:1 |
| 5[b] | Rh1 | AcOH | EtOH | <5 | – | – |
| 6[b] | Rh1 | AcOH | THF | 18 | 77 | 3:1 |
| 7[b] | Rh1 | AcOH | PhMe | <5 | – | – |
| 8[b] | Rh1 | AcOH | PhCl | 44 | 86 | 5:1 |
| 9[c] | Rh1 | AcOH | PhCF$_3$ | 25 | 89 | 8:1 |
| 10[c] | Rh1 | AcOH | Sol-1 | 57 | 88 | 6:1 |
| 11[c] | Rh1 | AcOH | Sol-2 | 52 | 83 | 10:1 |
| 12[c] | Rh1 | AcOH | Sol-3 | 40 | 90 | 12:1 |
| 13[d] | Rh1 | AcOH | Sol-4 | 47 | 91 | >20:1 |
| 14[d] | Rh1 | NaOAc | Sol-4 | 51 | 85 | >20:1 |
| 15[d] | Rh1 | PivOH | Sol-4 | 43 | 89 | >20:1 |
| 16[d] | Rh1 | 4-CF$_3$C$_6$H$_4$CO$_2$H | Sol-4 | 63 | 92 | >20:1 |
| 17[d,e] | Rh1 | 4-CF$_3$C$_6$H$_4$CO$_2$H | Sol-4 | 70 | 92 | >20:1 |
| 18 | Rh1 | 4-CF$_3$C$_6$H$_4$CO$_2$H | TFE | 78 | 64 | 1:2 |

[a]Reaction conditions A: amide **1** (0.1 mmol), alkyne **2** (0.1 mmol), (R)-Rh cat. (3 mol%), AgSbF$_6$ (12 mol%), AgOAc (2 equiv), and additive (1 equiv) in a solvent for 24 h, isolated yield.
[b]1 mL solvent, 12 h.
[c]1 mL solvent, 48 h.
[d]2 mL solvent, 72 h.
[e]60 °C. The e.e. was determined by HPLC using a chiral stationary phase.

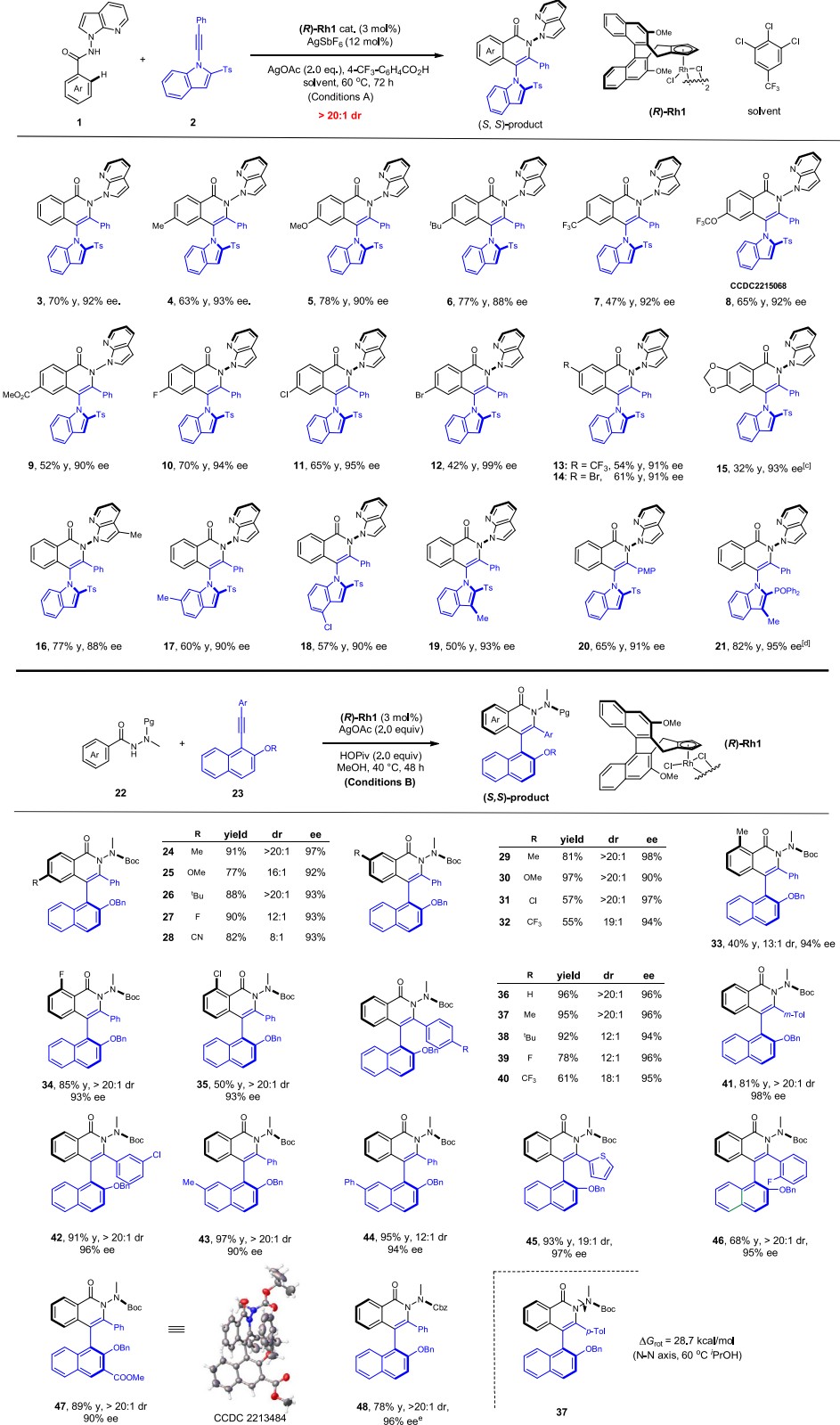

**Fig. 2 | Scope of the biaryl products with twofold chiral axes.** Reaction Conditions A: benzamide (0.1 mmol), alkyne (0.1 mmol), **(R)-Rh1** (3 mol%), AgSbF₆ (12 mol%), AgOAc (2 equiv), 4-CF₃C₆H₄COOH (1 equiv) at 60 °C in 1,2,3-trichloro-5-(trifluoromethyl)benzene (2 mL), 72 h, isolated yield; at 80 °C; in MeOH (1 mL).

Reaction Conditions B: benzamide (0.1 mmol), alkyne (0.12 mmol), **(R)-Rh1** (3 mol%), AgOAc (2.0 equiv) and HOPiv (2.0 equiv) in MeOH (2 mL), 40 °C, 48 h, isolated yield. Ag₂O (2.0 equiv), 30 °C.

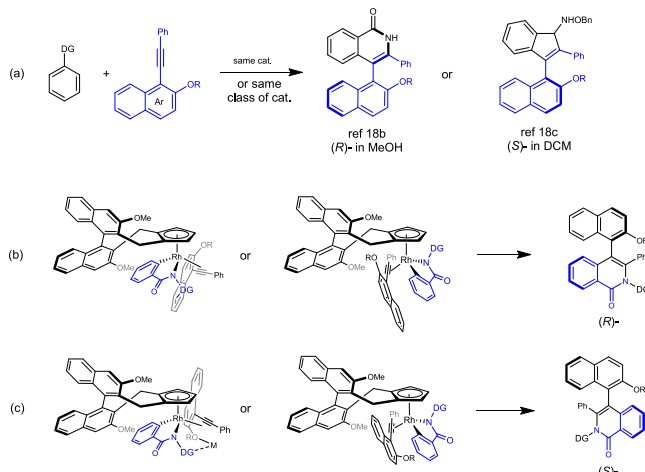

**Fig. 3 | Possible stereodivergence and chiral induction modes in enantioselective insertion of bulky alkynes. a** Our previous studies of Rh-catalyzed coupling of 1-alkynylnaphthalenes and arenes in [4 + 2] and [3 + 2] annulation. **b** Possible stereochemistry control via different orientations of the arene and alkyne. **c** Possible stereochemistry control via participation of additional metal.

presence of two rotamers in the NMR time-scale (along the N-Boc bond), and signal broadening was observed when the sample was slightly heated. In fact, removal of the Boc resulted in well-resolved NMR signals. The stereochemical stability of product **24** has been evaluated (Fig. 2). The fluxionality of the open-chain directing group resulted in reduced barrier of rotation along the N−N axis. Indeed, decreased diastereoselectivity was observed when it was heated at 60 °C, and the epimerization barrier was measured to be 28.7 kcal/mol. In contrast, the ee of the product remained essentially intact, indicating no C−C rotation.

During our previous studies of Rh-catalyzed coupling of 1-alkynylnaphthalenes and arenes[82,83], different configurations of the chiral axis were obtained even though a fixed (or fixed class of) chiral cyclopentadinenyl rhodium(III) catalyst was used (Fig. 3a). Although the arene reagents and the reaction patterns in Fig. 3a differed, the stereochemistry of the alkyne insertion seemed somehow related to the nature of the solvent; protic and aprotic solvents may operate with different trends. In this line, we previously proposed that in "innocent" solvents (DCE, DCM, PhMe), the Lewis acidic additive Ag(I) or Zn(II) may participate to bridge the directing group and the 2-OR group in the alkyne[84], resulting in a well-defined orientation of the alkyne toward insertion (Fig. 3). Indeed, during in our optimization studies of synthesis of 3, protic and aprotic solvent behaved differently. The employment of TFE solvent led to slight reversal of the diastereoselectivity (Table 1, entry 18), although the trend is insignificant. Although the observed oppositely configured C−C chiral axes in Fig. 3a refer to couplings using different classes of arenes, it is still possible in principle to develop diastereodivergent annulation starting from the same substrates by solvent control of the stereochemistry of the alkyne insertion (Fig. 3b, c). In order to maximize the stereodivergence, the steric bulk of the DG and the 2-OR group in the naphthalene ring of the alkyne must be suitable, and it should not always override other factors under different reaction conditions. In addition, the DG might have sufficient binding with an alcoholic solvent. These criteria might be satisfied when a planar N-directing group is adopted.

After extensive exploration of related benzamide substrates, we eventually realized diastereodivergence in the coupling of N-(7-azaindol-1-yl)benzamide (Fig. 4). As expected, the employment of DCE as a solvent in the presence of a halogenated benzoic acid allowed the synthesis of the [4 + 2] annulation product in ($S_{CC}$, $S_{NN}$) selectivity

(Conditions C). In contrast, the diastereomeric ($R_{CC}$, $S_{NN}$)-product was obtained when HFIP[91,92] was used as a strong hydrogen-bonding and polar solvent (Conditions D). Control experiments verified that the divergence was mainly dictated by the solvent although the acid additive differed. The scope of the diastereodivergence was then explored in detail. The DCE conditions were compatible with a broad scope of benzamides bearing a number of electron-donating and -withdrawing groups at the *ortho*, *meta*, and *para* positions of the benzene ring (**49–63**), and consistently excellent enantioselectivity was observed. Besides, the arene ring was also successfully expanded to a thienyl ring with excellent enantioselectivity (**64**, 95% ee). The combability of diversified substituents in the naphthalene ring or in the alkyne terminus was also ascertained (**66–73**), and the 2-substituent was not limited to OBn since 2-OTs group also allowed the desired coupling in excellent enantioselectivity (**74**, 94% ee). In all these cases, the couplings were highly diastereoselective (>20:1 dr). It was found that an appropriate oxygen group in the 2-position seems necessary to maintain catalytic activity. Thus, extension to 2-alkenyl, -methoxymethyl, -phenyl, and -OTf groups all failed to give any efficient coupling (<5% yield). These negative observations are in line with our mechanistic proposal in Fig. 3c in that the chelation between DG and the alkyne fragment is probably disrupted.

The scope of the diastereomerically complementary coupling was next explored under the HFIP conditions, which generally sustained more efficient coupling. Exactly the same arene substrate has been investigated under the Conditions D for direct comparisons. The reaction enantioselectivity echoed those under the DCE conditions ((*dia*)−**49–74**), and comparable or slightly lower enantioselectivity was observed in each specific case. Similarly, the diastereoselectivity was also slightly lower in some cases. Nevertheless, the conclusion of diastereodivergence of these two systems can be safely drawn. The advantageous side of the HFIP coupling conditions is the higher efficiency and better functional group compatibility as outlined in products (*dia*)-**75–80**, which failed to be obtainable under the DCE solvent conditions. A broader scope of the alkyne has been established, including those with an alkyl terminus ((*dia*)-**75**) and a 2,6-disubstituted bulky benzene ring ((*dia*)-**80**).

In each of the diastereoselective coupling systems, the reaction started with the initial construction of the N−N axis upon C−H activation (cyclometallation, see below and Fig. 4), followed by construction of the next C−C axis upon alkyne insertion. The C−N reductive elimination, however, does not seem to affect the N−N chirality. Thus, three selectivity parameters exist, and they were deconvoluted by following the Horeau principle (Kagan's treatment) by examination of the enantioselectivities of the major and minor diastereomers and the diastereomeric ratio (d.r.) (Fig. 5 and Supplementary Information). It was found that the selectivity parameter for the initial N−N chiral axis is $r_{CC}$ = 0.953 under the DCE conditions, which is very close to the ee (95.6%) of the major product and indicates that this process is enantiodetermining. Following the formation of the initial $S_{NN}$ (major) and $R_{NN}$ (minor) axis, the construction of the C−C axis then occurs with $r_1$ = 0.916 and $r_2$ = −0.780, and in the case of the initial minor $R_{NN}$ axis, a $R_{CC}$ axis is then induced. Nevertheless, the combination of $r_{NN}$ = 0.953 and $r_1$ = 0.916 allows the formation of the ($S_{NN}$, $S_{CC}$) diastereomer as the predominant product. A similar scenario was also observed under the conditions D in HFIP solvent, where the initial $S_{NN}$ axial was also favored with $r_{NN}$ = 0.934. The combination of this selectivity parameter and the $r_1$ = −0.938 led to the formation of the ($S_{NN}$, $R_{CC}$) diastereomer as the dominant product. In both cases, the construction of the C−C axis in the major product is predominantly influenced by the solvent.

## Computational mechanistic studies

To gain deep insight into the mechanism, we conducted computational studies to elucidate the mechanistic details (Fig. 6). According to the chiral induction mode proposed by Cramer[86–89] and experimentally

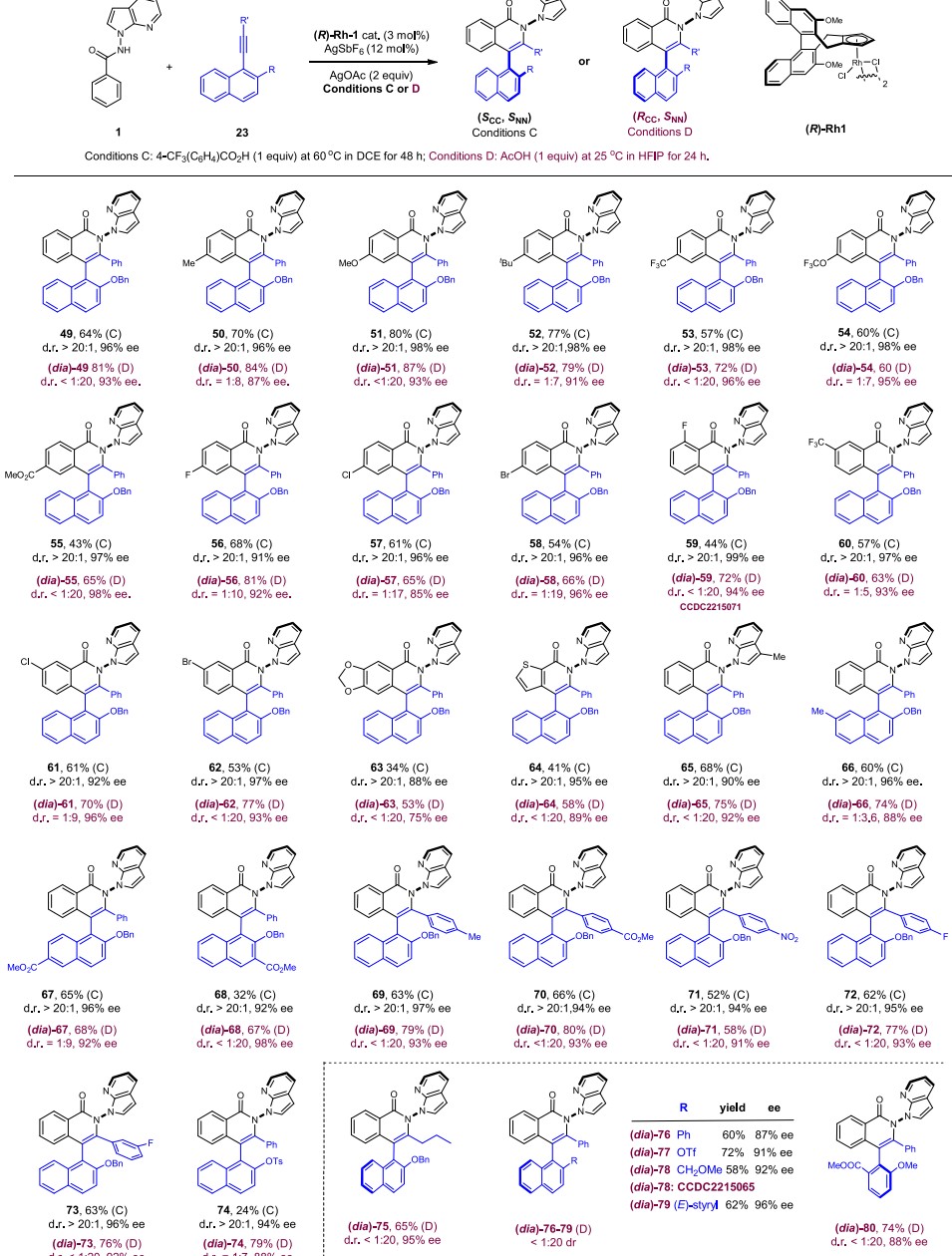

**Fig. 4 | Scope of diastereodivergent [4 + 2] annulation.** Reaction Conditions C: amide (0.1 mmol), alkyne (0.1 mmol), **(R)-Rh1** (3 mol%), AgSbF₆ (12 mol%), AgOAc (2 equiv), 4-CF₃-PhCOOH (1 equiv) at 60 °C in DCE (1 mL), 48 h, isolated yield. Reaction Conditions D: amide (0.1 mmol), alkyne (0.1 mmol), **(R)-Rh1** (3 mol%), AgSbF₆ (12 mol%), AgOAc (2 equiv), AcOH (1 equiv) at 25 °C in HFIP (1 mL), 24 h, isolated yield.

supported by our group, the C−H activation event generally occurred with a well-defined orientation of the arene substrate. Subsequently, the alkyne may approach with two different orientations (Fig. 5b, c), leading to the formation of C−C chiral axis with two opposite configurations during this enantio-determining insertion. Alternatively, You and coworkers proposed that the Rh−Cp bond may undergo facile rotation at the stage of a five-coordinate rhodacyclic intermediate, which changes the orientation of the cyclometalated arene[93]. The alkyne then approaches from a suitable side of the rhodacycle. Thus, the different combinations of orientations of the alkyne and the arene during the insertion would lead to differently configured C−C axis. In the coupling system, the solvent may play a key role in dictating the orientation of the alkyne or the mode or pattern of the alkyne

insertion. Detailed computational studies were then conducted to elaborate on the diastereodivergence (Fig. 6, and see Supplementary Figs. 122–127 and Supplementary Data 1). The computations reveal that the similar five-membered rhodacycles (S_NN)-**IM1** and (S_NN)-**IM2** can be generated via the C−H bond activation through the CMD mechanism in DCE and HFIP, respectively (see the Supplementary Fig. 122). Of particular note, it was found that the N−N axis rotation of (S_NN)-**IM1** and (S_NN)-**IM2** requires very high-energy barriers (see Supplementary Fig. 123 for details). Thus, the N−N axial chirality is determined upon C−H bond activation-cyclometalation, and the computed (S)-N−N axial chirality is indeed in accordance with the experiments.

In each solvent, the barriers of two key competing processes are evaluated theoretically, namely the direct insertion of the alkyne and

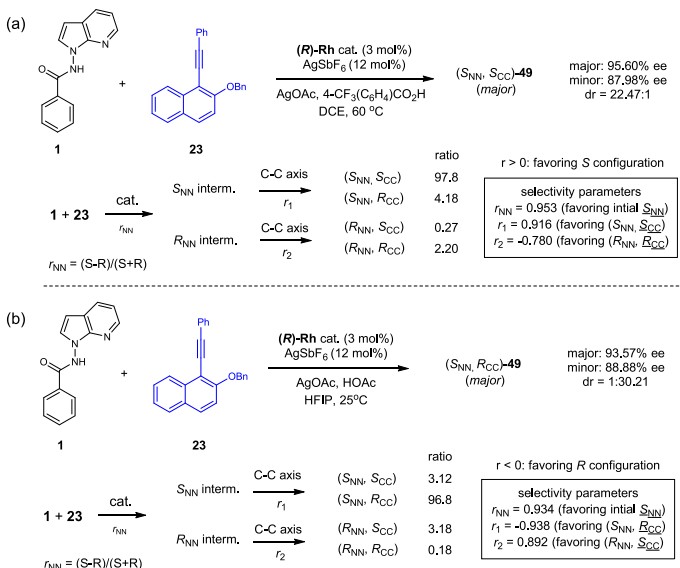

**Fig. 5 | Deconvolution of the three selectivity parameters.** Elucidations of three selectivity parameters in DCE (**a**) and HFIP (**b**).

the rotation along the Rh–Cp bond[93], which turns out to heavily affect the stereochemistry of the resulting C–C axis. For the reaction in DCE solvent (Fig. 6a), the barrier of the direct insertion of alkyne **23** into the Rh–C bond of ($S_{NN}$)-**IM1** was calculated to be higher. Instead, the five-coordinate ($S_{NN}$)-**IM1** prefers to undergo initial rotation along the Rh–Cp bond via transition state ($S_{NN}$)-**TS1** to give a diastereomeric rhodacycle ($S_{NN}$)-**IM1'**, and the barrier of this rotation (12.1 kcal/mol) is relatively. Although the resulting ($S_{NN}$)-**IM1'** species is higher in energy than the ($S_{NN}$)-**IM1**, the subsequent alkyne insertion was found to proceed with a lower barrier. Thus, in the two lowest energy pathways (Fig. 6a), the coordination of the C≡C triple bond of alkyne **23** in two different orientations leads to intermediates ($R_{CC}$, $S_{NN}$)-**IM3** and ($S_{CC}$, $S_{NN}$)-**IM3**. Then, the alkyne inserts via TSs ($R_{CC}$, $S_{NN}$)-**TS2** and ($S_{CC}$, $S_{NN}$)-**TS2**, giving rise to intermediates ($R_{CC}$, $S_{NN}$)-**IM4** and ($S_{CC}$, $S_{NN}$)-**IM4**, respectively, which would eventually furnish the [4 + 2] annulation products ($R_{CC}$, $S_{NN}$)-**49** and ($S_{CC}$, $S_{NN}$)-**49** after the C–N reductive elimination. It was calculated that the ($S_{CC}$, $S_{NN}$)-**TS2** is 1.8 kcal/mol lower in energy than ($R_{CC}$, $S_{NN}$)-**TS2**, which corresponds to a predicted d.r. of 15.2:1 at 60 °C, in qualitative agreement with the experimentally observed diastereoselectivity (>20:1 d.r.). When switching the solvent to HFIP (Fig. 6b), the hydrogen-bonding interaction between HFIP and O-atom of DG was observed, which can stabilize the five-membered rhodacycle and favors the corresponding alkyne insertion. It was found that the hydrogen-bonding interaction has little effect on the

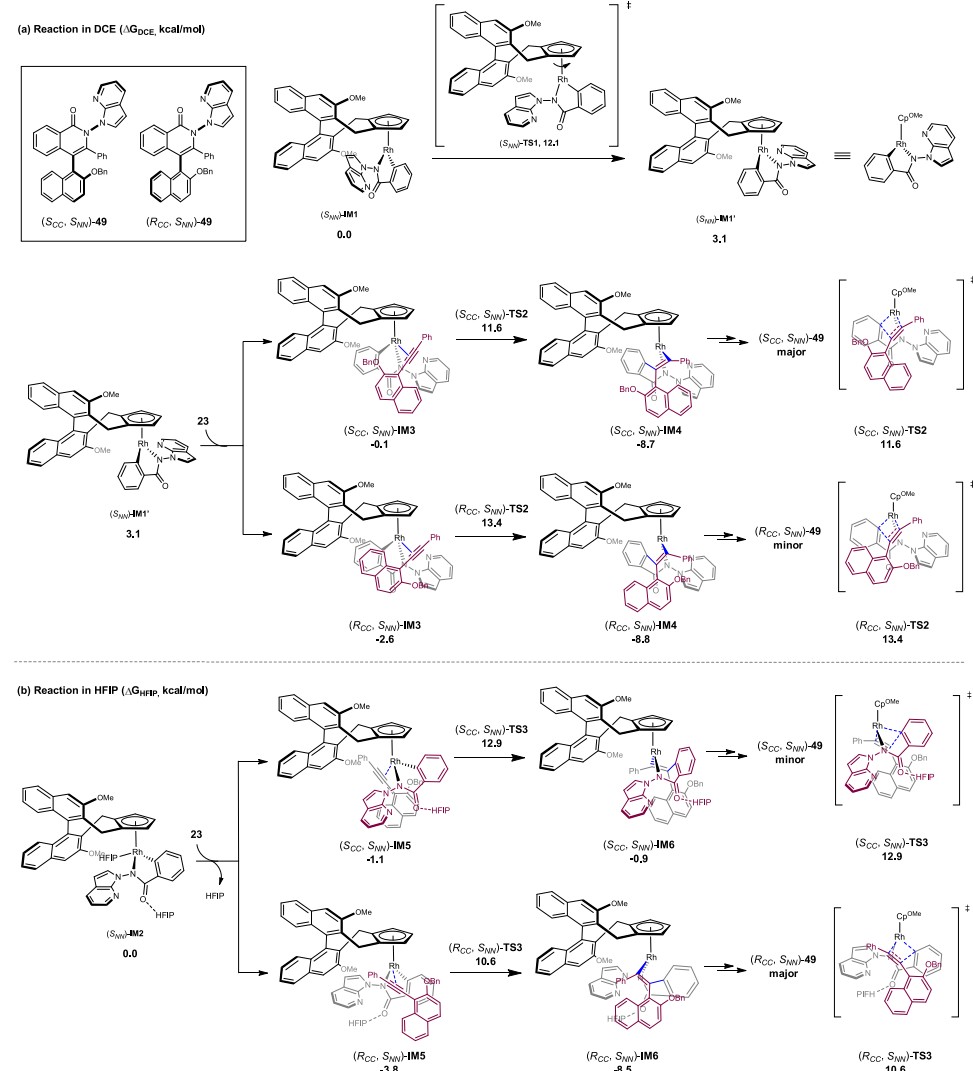

**Fig. 6 | Computational studies on the diastereodivergence during synthesis of product 49. a** Reaction in DCE ($\Delta G_{DCE}$, kcal/mol). **b** Reaction in HFIP ($\Delta G_{HFIP}$, kcal/mol).

**Fig. 7 | Reductive cleavage of the diastereomers toward enantiodivergence.**
Reaction conditions: (S,S) or (R, R) diastereomer (0.1 mmol), HMPA (100 µL), and
SmI₂ (25 equiv) in THF at −78 °C.

barrier of rotation of the rhodacycle around the Rh–Cp bond. However, the solvent effect is reflected by a lower barrier of the direct insertion of the alkyne when compared with that of the rotation along the Rh–Cp bond. Thus, the alkyne insertion was found to occur directly from the ($S_{NN}$)-**IM2** via the two lowest energy transition states ($R_{CC}$, $S_{NN}$)-**TS3** and ($S_{CC}$, $S_{NN}$)-**TS3**, which corresponds to the insertion of the alkyne from different sides of the rhodacycle. The energy of ($R_{CC}$, $S_{NN}$)-**TS3** is lower than that of ($S_{CC}$, $S_{NN}$)-**TS3** by 2.3 kcal/mol, which transforms to a predicted ratio of 48.5:1 at 25 °C, in line with a > 20:1 d.r. observed experimentally. In contrast, the insertion of the alkyne in DCE by duplicating the scenario depicted in the Fig. 6b only leads to higher energy barriers. Taken together, the computational outcomes indicate that the solvent has a significant impact on the mode of the alkyne insertion, modeling the experimentally observed diastereodivergence.

The possibilities of the ligation of Lewis acidic AgOAc or the hydrogen bonding with a carboxylic acid were also considered. As shown in Supplementary Fig. 128 (left), in the case of the above reaction in DCE (Conditions C), the transition states of alkyne insertion with participation of 4-CF₃C₆H₄COOH or AgOAc were found to be higher in energy compared to those devoid of such interactions. For the reaction in HFIP (Supplementary Fig. 128 right, Conditions D), our DFT studies indicated that the insertion transition states involving HFIP as the hydrogen bond donor are more favorable compared to those involving HOAc and AgOAc. These results suggest that the AgOAc and acid additives are unlikely to participate in this reaction step, and they should be readily substituted by HFIP. This is especially the case given the solvent nature of the HFIP which is in a large amount. These comparisons highlighted the uniqueness and effectiveness of HFIP in hydrogen bonding with the amide group.

To further support the conclusion of the chirality of the N–N axis, the amide substrate **1** was allowed to react with alkyne diphenylacetylene, a simpler alkyne, under our previous reaction conditions B, C, and D. The [4 + 2] annulation product ((S)-**81**) was obtained in 90% ee, 82% ee, and 77% ee, respectively, in good to excellent yields, with the same (S) configuration of the chiral N–N axis (Supplementary Fig. 120). Indeed, the employment of this simpler alkyne does not affect the chirality of the N–N axis. We also conducted racemization studies to determine the thermal stability of N–N axial chirality in **81**, and decay of enantioselectivity was observed when it was heated at 120 °C, giving a racemization barrier of 32.8 kcal/mol. In addition, to exclude the differences of temperature and reaction time, the coupling of amide **1** and alkyne **23** was conducted under modified conditions C and D at the same temperature for a fixed time (25 °C, 24 h). The corresponding products **49** or (dia)-**49** was each obtained with excellent enantioselectivity and diasteroselectivity, albeit with different efficiency.

The intrinsic oxidizing property of the N–N bond in the product renders the directing group cleavable under mild conditions. Treatment of a diaxes product with SmI₂ led to the rapid removal of the directing group, which left behind a single C–C chiral axis in the major product. By virtue of this simple operation, two diastereomeric

**Fig. 8 | Synthetic applications. a** (dia)-**49** (0.1 mmol), Br₂ (0.12 mmol), DCM (1 mL), 0 °C to rt, 12 h, isolated yield. **b** (dia)-**49** (0.1 mmol), m-CPBA (0.12 mmol), DCM (1 mL), 0 °C to rt, 12 h, isolated yield; **88** (0.1 mmol), 1-bromo-4-methylbenzene (0.1 mmol), Pd(OAc)₂ (5 mol%), Davephos (20 mol%), PivOH (30 mol%), Cs₂CO₃ (0.2 mmol), toluene (1 mL), 120 °C, 12 h, isolated yield. **c** (dia)-**77** (0.1 mmol), Ph₂P(O)H (0.4 mmol), Pd(OAc)₂ (10 mol%), dppb (10 mol%), NEt₃ (0.6 mmol), DMSO (1 mL), 110 °C, 12 h, isolated yield; **90** (0.1 mmol), HSiCl₃ (0.3 mmol), NEt₃ (0.6 mmol), toluene (1 mL), 100 °C, 12 h, isolated yield. **d** ethyl 2-diazo-2-phenylacetate (0.1 mmol), 1,2-dimethyl-1H-indole (0.1 mmol), Pd(PhCN)₂Cl₂ (10 mol%), **88** (10 mol%), NaBArF (24 mol%), DCM (1 mL), 25 °C, 12 h, isolated yield. **e** 1,3-diphenyl-2-propenyl acetate (0.1 mmol), dimethyl malonate (0.2 mmol), [Pd(allyl)Cl]₂ (2.5 mol%), **94** (6 mol%), K₃PO₄ (0.2 mmol), BSA (0.25 mmol), DCM (2 mL), 0 °C, 12 h, isolated yield.

products were converted to the corresponding enantiomers (Fig. 7). Thus, five pairs of such diastereomers have been readily converted to the enantiomers in excellent enantioselectivity (**82**–**86**). These products are known, and the absolute configurations were previously determined[82]. By following this two-step operation, the diastereodivergence has been transformed to enantiodivergence using the same chiral catalyst.

## Synthetic applications

Representative products have been studied toward synthetic or catalytic transformations as given in Fig. 8. Treatment of product (dia)-**49** with NBS afforded the 3-brominated product in excellent yield (**87**). The pyridine ring in **49** offers synthetic handles. m-CPBA oxidation of **49** afforded the N-oxide (**88**) in high yield. Further reaction with para-bromotoluene under palladium catalysis delivered the arylated product **89**. The OTf group in product (dia)-**77** was transformed to a phosphoryl group by palladium catalysis, and standard reduction of the phosphine oxide yielded product **91** as a potential chiral phosphine ligand. In all cases, essentially no erosion of the enantiopurity was detected. We next applied N-oxide **88** as a chiral ligand in Pd-catalyzed insertion of an N-protected indole into a donor–acceptor carbene reagent, affording **92** with promising enantioselectivity (59% ee)[73]. We also prepared a mono C–N axis phosphine ligand **94** via [4 + 2] annulation-reductive cleavage, and the allylic alkylation product **93** was obtained in 86% ee, indicating

that the axially chiral platform of the ligand played a key role in enantioselective control.

In summary, we report highly atroposelective synthesis of twofold axially chiral N–N biaryls in diversified reaction patterns via C–H activation of benzamides bearing an N-azaindolyl or open-chain N-directing group. The coupling with sterically hindered alkyne occurred as a result of dynamic kinetic transformation of both the benzamide and the alkyne substrates, affording three classes of diaxially chiral biaryl products in excellent enantioselectivity and high diastereoselectivity, which was enabled by the powerful and versatile role of the chiral Rh(III) catalyst that accommodates two independent stereo-determining steps. Most importantly, in addition to excellent enantioselectivity, rarely explored diastereodivergence was realized in the coupling of a class of benzamide and sterically hindered 1-alkynylnaphthalenes using the same catalyst. The origin of this diastereodivergence has been elaborated by DFT studies. Further simple cleavage of the N-directing group led to enantiodivergent synthesis of mono axially chiral C–C biaryls starting from the same substrates and chiral catalyst. The structural diversity of the N–N axially chiral platforms, excellent enantioselective control, and the diastreodivergence in this study may provide important insight into studies of complex axially chiral scaffolds through challenging asymmetric catalysis.

## Methods

### Synthesis of biaryl products with twofold chiral axes

**Conditions A**. A scew-cap vial (8 mL) was charged with N-(7-azaindol-1-yl)benzamide **1** (23.7 mg, 0.1 mmol, 1.0 equiv), 1-alkynylindole **2** (37.1 mg, 0.1 mmol, 1.0 equiv), **(R)-Rh1** (3.5 mg, 3 mol%), AgSbF$_6$ (4.2 mg, 12 mol%), AgOAc (33.4 mg, 0.2 mmol, 2.0 equiv), and 4-CF$_3$-C$_6$H$_4$CO$_2$H (19.0 mg, 0.1 mmol, 1.0 equiv). Trifluoromethyltrichlorobenzene solvent (2 mL) was then added, and the mixture was stirred at 60 °C for 72 h. The reaction mixture was evaporated under vacuum and the residue was purified by preparative TLC to give the corresponding product.

**Conditions B**. A scew-cap vial (8 mL) was charged with benzamides **22** (PG = Boc or Cbz, 0.1 mmol, 1.0 equiv), 2-substituted 1-alkynylnaphthalene **23** (0.12 mmol, 1.2 equiv), **(R)-Rh1** (3 mol%), AgOAc (0.2 mmol, 2.0 equiv), HOPiv (0.2 mmol, 2.0 equiv) and MeOH (2 mL) was then added, and the mixture was stirred at 40 °C for 48 h under air. The reaction mixture was evaporated under vacuum and the residue was purified by preparative TLC to give the corresponding product. The enantiomeric excess was determined by chiral HPLC analysis.

**Conditions C**. A scew-cap vial (8 mL) was charged with N-(7-azaindol-1-yl)benzamide **1** (23.7 mg, 0.1 mmol, 1.0 equiv), 2-substituted 1-alkynylnaphthalene **23** (0.1 mmol, 1.0 equiv), **(R)-Rh1** (3.5 mg, 3 mol%), AgSbF$_6$ (4.2 mg, 12 mol%), AgOAc (33.4 mg, 0.2 mmol, 2.0 equiv), and 4-CF$_3$-C$_6$H$_4$CO$_2$H (19.0 mg, 0.1 mmol, 1.0 equiv). DCE (1 mL) was then added, and the mixture was stirred at 60 °C for 48 h. The reaction mixture was evaporated under vacuum and the residue was purified by preparative TLC to give the corresponding product.

**Conditions D**. A scew-cap vial (8 mL) was charged with N-(7-azaindol-1-yl)benzamide **1** (23.7 mg, 0.1 mmol, 1.0 equiv), 2-substituted 1-alkynylnaphthalene **23** (37.1 mg, 0.1 mmol, 1.0 equiv), **(R)-Rh1** (3.5 mg, 3 mol%), AgSbF$_6$ (4.2 mg, 12 mol%), AgOAc (33.4 mg, 0.2 mmol, 2.0 equiv), and AcOH (6.0 mg, 0.1 mmol, 1.0 equiv). HFIP (1 mL) was then added and the mixture was stirred at 25 °C for 24 h. The reaction mixture was evaporated under vacuum and the residue was purified by preparative TLC to give the corresponding product.

## Data availability

The authors declare that the data supporting the findings of this study are available within the article and its Supplementary Information Files. The crystallographic data used in this study are available in the Cambridge Crystallographic Database under accession codes CCDC 2215068 (**8**), 2213484 (**47**), 2215071 ((*dia*)-**59**) and 2215065 ((*dia*)-**78**).

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

## Acknowledgements

The authors thank the Shandong University, Shaanxi Normal University, National Key R&D Program of China (Grant No. 2022YFA1503104 to X.L.), and National Natural Science Foundation of China (No. 22101167 to F.W. & No. 22073066 to G.H.) for financial supports.

## Author contributions

X.L. conceived the concept and directed the project. Y.W., X.Z., J.J., and F.W. conducted the experiments and data analysis. D.P. and G.H. conducted the DFT studies. X.L., G.H., and R.M. wrote the paper with feedback from all other authors.

## Competing interests

The authors declare no competing interests.
