## [Peer Review File · Nature Communications]

REVIEWER COMMENTS

Reviewer #1 (Remarks to the Author):

The manuscript by Li, Huang, and co-workers reports the synthesis of diverse classes of diaxially chiral biaryls containing N-N and C-N/C-C diaxes in distal positions through a rhodium-catalyzed enantioselective and diastereodivergent process. Despite extensive efforts have been performed for the synthesis of axially chiral scaffolds, the exploration of N-N axial chirality remains limited. This work represents a new strategy to achieve such a goal. Moreover, it provides a quite convenient access to diverse unique axially chiral scaffolds. Therefore, my feeling is that this work can be considered for publication in nat. commun. if these following questions could be addressed in a satisfactory manner with additional data.

(1) The author should reconsider the words “Diaxially Chiral Biaryls” in the title of this work. For example, the products 24-48 present in Table 2 look like a combination of axially chiral styrene and axially chiral hydrazine.

(2) Simple diphenylacetylene is suggested to use as an alternative in Scheme 4, and detect the enantiocontrol in the reaction of substrate 1 with diphenylacetylene under conditions B, C, and D. Such a reaction would be beneficial for mechanism studies. In principle, such a replacement would not affect the construction of N-N axial chirality. In this case, the authors are also suggested to conduct racemization studies to determine the thermal stability of N-N axial chirality. (From the Synthetic Applications in Scheme 8, seems very stable)

(3) Conditions C is operated at 60 oC for 48 h, while Conditions D is operated at 25 oC for 24 h, how about the results by operating Conditions C at 25 oC and Conditions D at 25 oC. (exclude the effect of temperature and reaction time)

(4) In the proposed mechanism and DFT calculation, the author declares that the HFIP acts as a hydrogen bond donor and ligand in the catalytic system, and changes the sequence of two competing elementary steps. However, the 1.0 equivalent of 4-CF₃-C₆H₄COOH is used in conditions C, and such an acid might be more strong hydrogen bond donor than HFIP, thus the author should consider the role of 4-CF₃-C₆H₄COOH during mechanism studies and DFT calculations (Scheme 6).

(5) The 1.0 equivalent of AcOH used in conditions D should be also considered in the mechanism studies.

Other minor modifications required:

1. use Supplementary Table 1,2,3.... or Supplementary Figure 1,2,3....to replace the words “Supporting Information” in main text

Reviewer #2 (Remarks to the Author):

Li and co-worker report here a novel rhodium-catalyzed atroposelective synthesis of chiral biaryls with both C-N and N-N axial chiralities in a selective and divergent way. Recently, the construction of axial chirality become a hot research topic in organic synthesis due to the important role of such compounds in asymmetric catalysis and the field of pharmaceutical research. Compared with the plenty of methods for axial C-C bond formation, the methods for axial N-N bond are limited. Since 2021, there are two general methods for the direct construction of axial N-N bond. The first is ring-formation and the second is desymmetry. Now, Li and co-workers describe here the use of asymmetric C-H bond activation to achieve this goal. Moreover, this method can forge axial C-N and N-N bonds, or C-C and N-N bonds concurrently with excellent diastereoselectivity. The substrate scope is great and lots of products have been prepared. The data is convinced and the manuscript has been well prepared. Overall, acceptance is strongly recommended without the requirement of further revision.

Reviewer #3 (Remarks to the Author):

The mechanistic study by Huang, Li and coworkers tries to investigate an interesting mechanism involving both N-N and C-C axial chirality. They have used DFT methods to primarily understand the diastereodivergence in two different solvents, namely, DCE and HFIP. While the chosen methodology and mechanism are reasonable, the control of diastereodivergence via a H-bonding interaction with HFIP seems a bit far-fetched currently. The main concern here is the presence of the oxidant AgOAc and additives; trifluoromethylbenzoic acid/acetic acid. What is the role of these additives? It is very likely that these additives can also H-bond similar to HFIP. In that case, how is the H-bonding of HFIP alone resulting in a different diastereomer? Have the authors tried including trifluoromethylbenzoic acid explicitly in the TSs; (SCC, SNN)-TS2 and (RCC, SNN)-TS2? Will that result in a model similar to the one proposed with HFIP? Apart from this, the authors mention the role of Lewis acids in line 147. Have they tried including AgOAc in the TS model?

With the presence of AgOAc and other acids, the H-bonding with HFIP alone seems doubtful. This requires a more detailed study before the exact role of solvents can be elucidated in diastereodivergence.

Reviewer #1 (Remarks to the Author):

The manuscript by Li, Huang, and co-workers reports the synthesis of diverse classes of diaxially chiral biaryls containing N-N and C-N/C-C diaxes in distal positions through a rhodium-catalyzed enantioselective and diastereodivergent process. Despite extensive efforts have been performed for the synthesis of axially chiral scaffolds, the exploration of N-N axial chirality remains limited. This work represents a new strategy to achieve such a goal. Moreover, it provides a quite convenient access to diverse unique axially chiral scaffolds. Therefore, my feeling is that this work can be considered for publication in nat. commun. if these following questions could be addressed in a satisfactory manner with additional data.

(1) The author should reconsider the words “Diaxially Chiral Biaryls” in the title of this work. For example, the products 24-48 present in Table 2 look like a combination of axially chiral styrene and axially chiral hydrazine.

Reply: “Diaxially Chiral Biaryls” has been changed to Diaxially Chiral Heterocycles.

(2) Simple diphenylacetylene is suggested to use as an alternative in Scheme 4, and detect the enantiocontrol in the reaction of substrate **1** with diphenylacetylene under conditions B, C, and D. Such a reaction would be beneficial for mechanism studies. In principle, such a replacement would not affect the construction of N-N axial chirality. In this case, the authors are also suggested to conduct racemization studies to determine the thermal stability of N-N axial chirality. (From the Synthetic Applications in Scheme 8, seems very stable)

Reply: We thank the referee for pointing out this. We have applied diphenylacetylene as the coupling reagent to detected the enantiocontrol in the reaction of substrate **1** under conditions B, C, and D. The ee's of the product (*S*-**81**) were found to be 90%, 82%, 77%, respectively. Indeed, the employment of this simpler alkyne really does not affect the chirality of the N-N axis. We also conducted racemization studies to determine the thermal stability of N-N axial chirality in (*S*)-**81**, and decay of enantioselectivity was observed when it was heated at 120 °C (mesitylene), giving a racemization barrier of 32.8 kcal/mol (Scheme 8).

(3) Conditions C is operated at 60 °C for 48 h, while Conditions D is operated at 25 °C for 24 h, how about the results by operating Conditions C at 25 °C and Conditions D at 25 °C. (exclude the effect of temperature and reaction time)

Reply: We thank the referee for pointing out this. In order to exclude the effect of temperature and reaction time, the reaction was conducted under Conditions C but at 25 °C for 24 h, and the product **49** was obtained in 23% yield and 97% ee with excellent diastereoselective (> 20:1 dr). The reaction in Conditions D was operated at 25 °C for 24 h, and the product (*dia*)-**49** was obtained in 81% yield and 93% ee with excellent diastereoselective (> 20:1 dr). These observations reinforced our conclusions (Scheme 8).

(4) In the proposed mechanism and DFT calculation, the author declares that the HFIP acts as a hydrogen bond donor and ligand in the catalytic system, and changes the sequence of two competing elementary steps. However, the 1.0 equivalent of 4-CF₃-C₆H₄COOH is used in conditions C, and such an acid might be more strong hydrogen bond donor than HFIP, thus the author should consider the role of 4-CF₃-C₆H₄COOH during mechanism studies and DFT calculations (Scheme 6).

Reply: We thank the referee for raising this concern. We have considered the possibility of 4-CF₃-C₆H₄COOH acting as a hydrogen bond donor in our DFT calculations. However, the results showed that in the case of the reaction in DCE (Conditions C), the insertion transition states that involve such a hydrogen bond with 4-CF₃-C₆H₄COOH are higher in energy compared to those that do not involve hydrogen bonding (entropic effect). This suggests that 4-CF₃-C₆H₄COOH is unlikely to participate in this reaction step (Scheme 7). For more details, please refer to our responses to referee #3.

(5) The 1.0 equivalent of AcOH used in conditions D should be also considered in the mechanism studies.

Reply:

Reply: We thank the referee for pointing out this. Besides HFIP, HOAc was also considered as the hydrogen bond donor and was evaluated for the reaction in HFIP (Conditions D). The results indicated that the insertion transition states involving HFIP as the hydrogen bond donor are more favorable compared to those involving HOAc. For more details, please refer to our response to referee #3 (Scheme 7).

Other minor modifications required:

1. use Supplementary Table 1, 2, 3... or Supplementary Figure 1, 2, 3... to replace the words "Supporting Information" in main text

Reply: We have corrected the formats.

Reviewer #2 (Remarks to the Author):

Li and co-worker report here a novel rhodium-catalyzed atroposelective synthesis of chiral biaryls with both C-N and N-N axial chiralities in a selective and divergent way. Recently, the construction of axial chirality become a hot research topic in organic synthesis due to the important role of such compounds in asymmetric catalysis and the field of pharmaceutical research. Compared with the plenty of methods for axial C-C bond formation, the methods for axial N-N bond are limited. Since 2021, there are two general methods for the direct construction of axial N-N bond. The first is ring-formation and the second is desymmetry. Now, Li and co-workers describe here the use of asymmetric C-H bond activation to achieve this goal. Moreover, this method can forge axial C-N and N-N bonds, or C-C and N-N bonds concurrently with

excellent diastereoselectivity. The substrate scope is great and lots of products have been prepared. The data is convinced and the manuscript has been well prepared. Overall, acceptance is strongly recommended without the requirement of further revision.

Reply: We thank the referee for the justifiable positive comments.

Reviewer #3 (Remarks to the Author):

The mechanistic study by Huang, Li and coworkers tries to investigate an interesting mechanism involving both N-N and C-C axial chirality. They have used DFT methods to primarily understand the diastereodivergence in two different solvents, namely, DCE and HFIP. While the chosen methodology and mechanism are reasonable, the control of diastereodivergence via a H-bonding interaction with HFIP seems a bit far-fetched currently. The main concern here is the presence of the oxidant AgOAc and additives; trifluoromethylbenzoic acid/acetic acid. What is the role of these additives? It is very likely that these additives can also H-bond similar to HFIP. In that case, how is the H-bonding of HFIP alone resulting in a different diastereomer? Have the authors tried including trifluoromethylbenzoic acid explicitly in the TSs; (SCC, SNN)-TS2 and (RCC, SNN)-TS2? Will that result in a model similar to the one proposed with HFIP? Apart from this, the authors mention the role of Lewis acids in line 147. Have they tried including AgOAc in the TS model?

With the presence of AgOAc and other acids, the H-bonding with HFIP alone seems doubtful. This requires a more detailed study before the exact role of solvents can be elucidated in diastereodivergence.

Reply: We thank the referee for pointing out this. The possibilities of the ligation of Lewis acidic AgOAc and hydrogen bonding with a carboxylic acid were considered. As shown below, in the case of the reaction in DCE (Conditions C), the insertion transition states that involve interactions with 4-CF₃C₆H₄COOH or AgOAc are all higher in energy compared to those that devoid of such interactions. For the reaction in HFIP (Conditions D), our DFT studies also indicate that the insertion transition states involving HFIP as the hydrogen bond donor are more favorable compared to those involving HOAc and AgOAc. This results suggest that the AgOAc and acid additives are unlikely to participate in this reaction step, and their binding should be readily substituted by HFIP. This is especially the case given the solvent nature of the HFIP which is in a large amount. Our studies highlighted the uniqueness and effectiveness of HFIP in hydrogen bonding and reinforced our conclusion. These information has been added to Scheme 7.

(a) Reaction in DCE (ΔG_{DCE} , kcal/mol)

(S_{CC} , S_{NN})-TS2
11.6

(R_{CC} , S_{NN})-TS2
13.4

acid = 4-CF₃-C₆H₄COOH

(S_{CC} , S_{NN})-TS2'
12.0

(R_{CC} , S_{NN})-TS2'
14.4

(S_{CC} , S_{NN})-TS2''
14.8

(R_{CC} , S_{NN})-TS2''
17.5

(b) Reaction in HFIP (ΔG_{HFIP} , kcal/mol)

(S_{CC} , S_{NN})-TS3
12.9

(R_{CC} , S_{NN})-TS3
10.6

(S_{CC} , S_{NN})-TS3'
18.2

(R_{CC} , S_{NN})-TS3'
14.3

acid = HOAc

(S_{CC} , S_{NN})-TS3''
19.0

(R_{CC} , S_{NN})-TS3''
13.9

REVIEWERS' COMMENTS

Reviewer #1 (Remarks to the Author):

The authors have done a good job in replying to my comments. I gladly recommend this work for publication in Nature Communication.

Reviewer #3 (Remarks to the Author):

The authors have addressed all comments and the manuscript can now be accepted.